# Experimental Study on a Prefabricated Lightweight Concrete-Filled Steel Tubular Framework Composite Slab Structure Subjected to Reversed Cyclic Loading

**Jia Suizi [1], Cao Wanlin [2,\*], Liu Zibin [3], Ding Wei [1] and Su Yingnan [1]**

1   School of Engineering and Technology, China University of Geosciences, Beijing 100083, China;
    2016010023@cugb.edu.cn (J.S.); 2102180030@cugb.edu.cn (D.W.); 2102180031@cugb.edu.cn (S.Y.)
2   College of Architecture and Civil Engineering, Beijing University of Technology, Beijing 100124, China
3   Shandong Provincial Architectural Design Institute, Jinan city, Shandong province, Qindao 250001, China;
    Abby2011@126.com
*   Correspondence: 07814@bjut.edu.cn; Tel.: +86-151-2007-4483; Fax: +86-10-6739-6617

**Abstract:** A building structure comprising a prefabricated lightweight concrete-filled steel tubular (CFST) framework composite slab structure is proposed. Five full-scale specimens (i.e., one empty framework and four-walled frameworks) were tested under reversed cyclic loading to study their earthquake-resistance performance. Of the four wall specimens, three were walled using composite slabs, one had no openings, one had a window opening, and one had a door opening. One was walled with a concealed steel-truss slab. A comparative study on the strength, stiffness, ductility, hysteresis characteristics, and dissipated energy of the specimens was performed. The working mechanism of the framework and slab was then analyzed. The results show that, if reasonably assembled and connected, the framework and slab work in a well-coordinated manner. The walled framework had greater lateral load-bearing capacity, better energy-dissipation, greater stiffness reduction, and better deformability than an empty framework. The area and type of slab opening had a significant impact on structural performance because a door or window opening contributed to a smaller lateral load-bearing capacity and initial secant stiffness of the structure. However, this had no clear impact on the accumulative dissipated energy of the structure.

**Keywords:** prefabricated lightweight structure; concrete-filled steel tubular framework; composite slab; reversed cyclic loading; concealed steel-truss slab

---

## 1. Introduction

In China, many rural area homes enduring cold or very cold weather are generally heated with simple crude equipment. In the winter, energy consumption for heating purposes accounts for approximately 80% of the total domestic energy consumption. This results in only a massive waste but also an uncomfortable indoor environment. In rural regions with hot summers and cold winters, heating equipment is seldom used. The walls and enveloping structures of rural buildings tend to have a low thermal efficiency and the average indoor temperature of rural homes in the winter is much lower than that of urban homes. In rural areas with hot summers, electric fans and air conditioners have gained popularity due to economic development and improvements of living standards. However, the poor heat-insulation performance of rural buildings results in a drastic increase in electricity consumption. Therefore, energy conservation is a significant challenge for rural residence construction. Additionally, many rural homes are built in earthquake regions. The majority are self-built using construction technologies that cannot meet the basic requirements for earthquake resistance. This results in buildings with poor earthquake-resistance. Clearly, it is a pressing matter

to develop technologies for low-rise rural buildings that can meet requirements of both earthquake resistance and energy savings [1,2].

Compared to traditional rural building structures, concrete-filled steel tubular (CFST) framework structures and steel structures have distinct advantages. Steel structures have the advantages of light self-weight, high-strength, and good deformability. They, therefore, have good earthquake resistance. Additionally, steel structures can be quickly fabricated to facilitate rapid construction in a highly industrialized manner. Furthermore, steel structures are a greener solution because they do not produce construction waste when dismantled. CFST structures not only have the advantages of steel, they also improve fire-resistance and durability and extend the life cycle of building structures, which produces socio-economic benefits [3,4].

Wall structures using heat-insulation elements can solve the issue of poor heat in rural buildings. An increasing level of industrialization in the housing industry means that there will be an increasing number of new walls with good energy-savings and heat-insulation performance. New energy-saving and heat-insulating walls, with steel structures or CFST structures, can effectively improve the heat-insulation performance of building structures.

Both steel and CFST structures are already widely used in high-rise buildings. However, due to their high cost, they are seldom applied in rural buildings. The steel industry in China suffers industry-wide losses because of increased outputs with decreasing demands. Thus, it is a critical challenge to address the issue of excess capacity. The application of lightweight steel materials in rural buildings could not only improve the earthquake-resistance performance of rural buildings, it could also reduce the excess capacity of the industry. This provides an opportunity to utilize steel or CFST structures in the construction of rural buildings.

In China, industrialized building construction technologies are mainly used for multi-storey and high-rise residential buildings. In contrast, there is no clear distinction between research related to urban and rural buildings in developed countries. In some developed countries, one-story and two-story buildings in rural and suburban areas are all constructed using industrialized technologies. These buildings differ from the traditional building structures of China's rural areas and do not fit the rural economic statuses of the nation. It should be noted that there is inadequate research into the improvement of human settlement and the realization of sustainable new community development and industrialized building construction in rural China.

The development of new precast structures has always been a key interest for researchers in the field of structural engineering, which is the subject of extensive experimental research and theoretical analysis.

The seismic behavior of floor diaphragms and pinned beam–column connections in a multi-storey precast building has been addressed experimentally. Dry mechanical connections were adopted to realize the joints between floor-to-floor, floor-to-beam, wall-to-structure, column (and wall)-to-foundation, and beam-to-column. The results demonstrated that the proposed new beam-to-column connection system was a viable solution for enhancing the response of precast reinforced-concrete (RC) frames subject to seismic load [5]. A novel prestressed precast reinforced concrete beam-to-column joint was presented. The beam ends adjacent to the column were reinforced with a steel jacket to prevent concrete from spalling. Steel strands were used to provide the joint with a self-centering capacity. Replaceable mild steel bars provided the joint with an energy dissipation capacity. The effectiveness of the proposed joint was investigated via experimental tests. The obtained results demonstrated the beneficial properties of the joint in terms of strength and deformation capacity, control of story drift, easy reparability, and prevention of beams and columns from damage [6]. Two precast and two monolithic concrete joints for exterior beam-to-column connections were tested under cyclic loading. The installation of precast specimens was prepared using the dry-type method, whereas the monolithic joints were cast in situ. The results showed that the precast joints performed a satisfactory resistance to seismic loads [7].

To enhance the earthquake resistance of emulative precast concrete (PC) panels, two potential methods were studied. First, by using the reduced rebar area, the PC panel at the wall bottom was weakened to develop a plastic hinge zone in the PC panel rather than at the panel joints. Second, a hybrid wall was considered, where cast-in-place concrete was used for the plastic hinge zone at the bottom panel. Four specimens, including an ordinary RC wall, were tested under cyclic loading. The test results revealed that gap opening and shear slip at the panel joints were prevented by using the proposed methods [8]. The lateral load behavior of two, 0.40-scale hybrid precast concrete shear wall-test specimens was evaluated to emulate monolithic cast-in-place RC shear walls. The results demonstrated the potential for the use of precast walls in seismic regions [9]. The seismic performance of the precast structure of a frame-supported multi-ribbed composite wall with a large space at the bottom was studied under cyclic loading. The results showed that the structure, as a whole, provided extraordinary seismic performance because of the unique layer-by-layer embedded design of the wall [10–12]. Experimental and analytical studies were performed to examine the structural behavior of precast foam concrete sandwich panels (PFCSP) under vertical in-plane shear loads. Six full-scale PFCSP specimens, varying in heights, were developed to study an important parameter such as the slenderness ratio (H/t). The results showed that the PFCSP wall was a potential alternative to the conventional load-bearing wall system [13]. A 1/3-scale model of a five-story self-centering precast reinforced concrete frame with shear walls was designed and tested on a shake-table under a series of bi-directional earthquake excitations of increasing intensity. The results demonstrated that seismic performance of the test specimen was satisfactory in the plane of the shear wall [14]. Experimental and numerical results was presented for full-scale Blockhaus shear walls subjected to in-plane lateral loads. It was interesting to notice that this deformation limit largely exceeds the ultimate lateral displacement of light-frame shear walls or masonry filled timber-framed walls under in-plane lateral loads [15]. The non-linear modelling of the cyclic behavior of Blockhaus timber log-walls under in-plane lateral loads was investigated. The behavior of Blockhaus log-walls under lateral loads markedly depended on their corner joints. Pinching behavior, strength, and stiffness degradation in joints were properly described [16].

Steel frames with reinforced concrete infill walls (SRCWs) were an interesting seismic-resistant structural solution. The seismic behavior of SRCWs had been the object of many theoretical and experimental studies.

Hybrid steel and concrete systems made by SRCWs due to their many advantages as seismic-resistant systems [17,18], i.e., high initial stiffness beneficial in reducing building damage under low-intensity earthquakes, effective damping characteristics, and potentially easy repairs after moderate damage through the use of epoxy resins on the cracked wall. In order to investigate the behavior of partially-restrained steel frame with RC infill wall (PSRCW), two specimens with one-third scale, one-bay, and two-story were performed under a reversed cyclic lateral load. Test results showed that PSRCW with solid infill walls exhibited moderate ductility capacity and energy dissipation due to the degradation of post-peak strength. PSRCW with concealed vertical slits exhibited much larger ductility, deformability, and energy dissipation capacity than the other one [19]. A study was performed to investigate concrete masonry infills bounded by steel frames and then used the resulting data to evaluate the effectiveness of equations, which suggested the current design standards for both Canada and the US for the design of masonry infills [20]. Shaking table tests on steel frames with autoclaved lightweight concrete external walls were studied. The results showed the presence of infill had obvious impact on the stiffness and load-carrying capacity of the combined systems [21]. An innovative CFS wall with infilled lightweight flue gas desulfurization (FGD) gypsum was presented in this paper. Test observations showed that the infilled gypsum behaved as a diagonal bracing along with wall frames to resist shearing forces [22].

Four steel frame with infill walls specimens were tested under horizontal cyclic loadings. Both experimental and analytical results showed that the stiff RC infill wall dominated the lateral resistance and drift capacity of the test specimens, and that, by adding slit-separated features at the

edges of infill walls, improved the drift capacity [23]. In-plane seismic behavior of concrete sandwich panel-infilled steel frame (CSP-ISF) was experimentally and numerically investigated.

It was found that addition of sandwich panels leads to considerable increase in the lateral stiffness and strength, ductility, energy dissipation capacity, and equivalent viscous damping ratio of the steel frames [24].

The following deficiencies were identified in existing studies.

(1) Traditional CFST structures are mainly used for multi-story and high-rise buildings. Few are suitable for low-rise buildings and even fewer are suitable for industrialized construction of low-rise buildings.

(2) Currently, the beam and column of a concrete-filled steel-tube framework are usually connected by a structure comprising a reinforcing ring in the outer side and a diaphragm in the inner side. This joint structure requires a complicated construction process, and the prefabrication level is low.

(3) Extant studies on sandwich slabs have focused on the mechanical properties and earthquake-resistance performance of the walls, with scant investigation on the earthquake-resistance performance of walls mounted in frameworks.

In this study, a prefabricated concrete-filled steel tubular (CFST) framework composite slab structure is proposed to address the issues associated with rural buildings in China and the deficiencies of extant studies. This type of structure is mainly suitable for low-rise rural buildings and is intended to improve earthquake-resistance and energy-saving performance. The beam and column of the light-weight steel framework are connected by a structure comprising double stiffener-reinforced L-brackets that can easily be prefabricated. The framework can be walled within two types of composite slabs including a sandwich slab and a concealed steel-truss slab. The sandwich slab has an integrated heat-insulation element. The concealed steel-truss slab can improve the lateral load-resistance capacity of the lightweight steel framework. A comparative analysis of the strength, stiffness, hysteresis characteristics, energy dissipation capacity, and the failure mechanism of the specimens is performed.

Considering the durability, heat preservation, fire resistance, and impact resistance of the wall, the sandwich composite wall is used in the cross-sectional construction of this wall. The main types and characteristics are described next.

The first type of composite wall is the mortar sandwich layer, which is a polystyrene granular mortar layer, while the two side layers are ordinary mortar surfaces with steel wires. The surface thicknesses are approximately equal to 20 mm and the strength of the surface layer can attain values >5 MPa. These characteristics meet the requirements of the impact resistance of the surface layer. Based on the condition of equal thickness, the weight of the wall is lighter, but the insulation effect in the sandwich layer of the polystyrene granular mortar is not as good as that of the polystyrene board.

The second type relates to the fact that the composite wall is formed by a polystyrene board sandwich layer together with fine stone concrete surfaces and steel wire meshes on both sides. Accordingly, the thickness of the surface layer is approximately 50 mm, and the strength of the surface layer can attain values >20 MPa, which can meet the requirements of the impact resistance of the surface layer. However, the weight of the wall is heavier than that of the first type if the condition of equal thickness is assumed to be valid.

The third type refers to the composite slab developed in this study, whereby the middle layer is an insulation layer of graphite polystyrene board, and the two side layers are high-performance foam concrete structural layers with steel wire meshes. The surface thickness is in the range of 50 to 80 mm and the surface strength can attain values >5 MPa. Accordingly, these characteristics meet the impact resistance requirements of the surface layer. Additionally, the weight of the wall is equivalent to that of the first type if the condition of equal thickness, insulation effect, and fire resistance of the wall are better than those of the former types.

## 2. Experimental Review of Prefabricated Lightweight CFST Framework-Composite Slab Structures

### 2.1. Experimental Design and Fabrication of Specimens

Five full-scale specimens of prefabricated lightweight CFST framework composite slab structures were developed for this study.

#### 2.1.1. Design of the Framework

The framework comprised lightweight recycled CFST columns and steel H-beams joined with a structure of double stiffener-reinforced L-brackets. The specification of the square steel tube for the columns was 150 × 150 × 6 mm. The specification of the steel H-beam was HM 194 × 150 × 6 × 9 mm. To join the column and beam, two stiffener-reinforced L-brackets were welded to the column and connected to the beam with M12 high-strength bolts (grade 8.8), as shown in Figure 1.

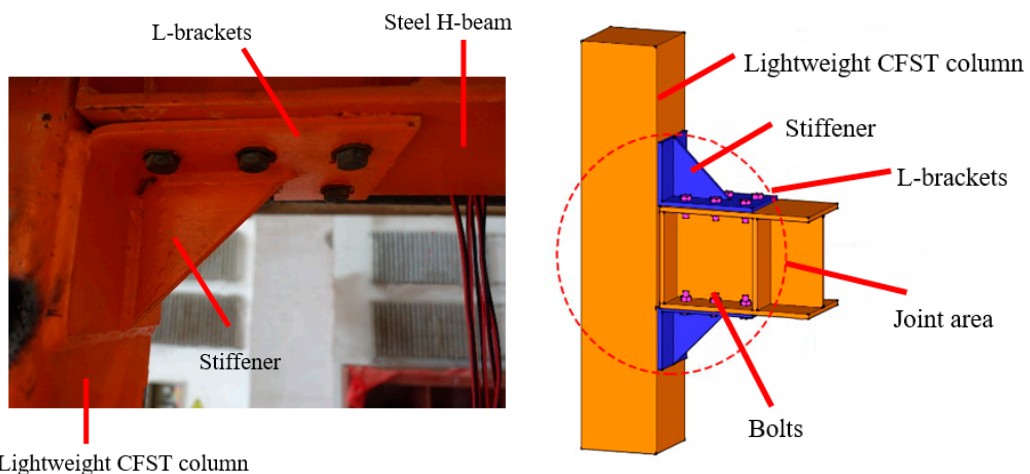

**Figure 1.** Image and figure showing stiffener-reinforced L-brackets welded to the column and connected to the beam with M12 high-strength bolts.

#### 2.1.2. Design of a Composite Slab

Two types of composite slab were prepared for the experiment including a sandwich panel and a concealed steel-truss panel. Three designs of the sandwich slab were prepared without an opening, with a window opening, and with a door opening. The specimens were designed with a column spacing of 3900 mm and a floor height of 2700 mm. The concealed steel-truss and sandwich panels were designed with thicknesses of 150 and 240 mm, respectively.

Sandwich panels were spliced together to form a slab without an opening. The sandwich panel measured 590, 3540, and 240 mm in height, length, and thickness, respectively. It comprised three layers with each measuring 80 mm in thickness. The dimensions of the sandwich panel could be changed, according to the design of the column spacing and floor height. Thus, it was suitable for different spatial layout designs of rural buildings.

The middle layer of the sandwich panel was a graphite-polystyrene board, serving as the heat-insulation layer of the slab, whereas the outer layers comprised cast-foamed concrete and served as the structural layers. The thickness of the middle heat-insulation layer can be increased or decreased, according to the different energy-saving building standards and the requirements of different regions. Additionally, foamed concrete layers have a smaller thermal conductivity than normal concrete and masonry blocks, which contributes to the improved heat-insulation performance of the slab. The outer structural layers of foamed concrete also protect and improve the fire-resistance and durability of the middle heat-insulation layer. A single layer of φ3@50 orthogonal galvanised cold-drawn steel wire mesh with 30-mm cover thickness was embedded into each of the outer structural layers of foamed concrete to improve strength. The wire meshes in the two structural layers were connected with φ3

wires diagonally embedded across the heat-insulation layer, which form a spatial wire truss framework that enabled the outer structural layers and middle heat-insulation layer to form an integrated unit for load bearing.

A steel-tie bar of φ6@200 was attached to each end of the panel for connection with the columns of the framework. The structure was partly embedded into the panel with a buried depth of 300 mm and exposed a straight length of 105 mm. It was welded with the single layer of the orthogonal galvanized cold-drawn steel wire mesh using a minimum of three welding points. Two sandwich panels were connected with a tongue-and-groove joint. The bottom edge of the panel was of a groove design and the top edge was a tongue design. Figure 2 showed the structural design of the sandwich panel.

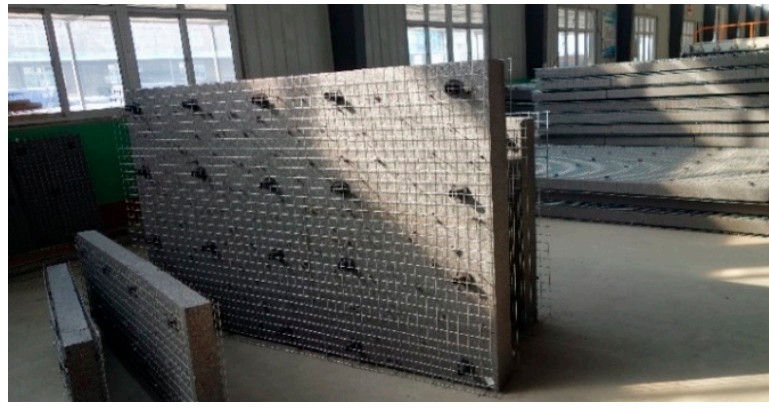

(**a**) Structural design of the sandwich panel

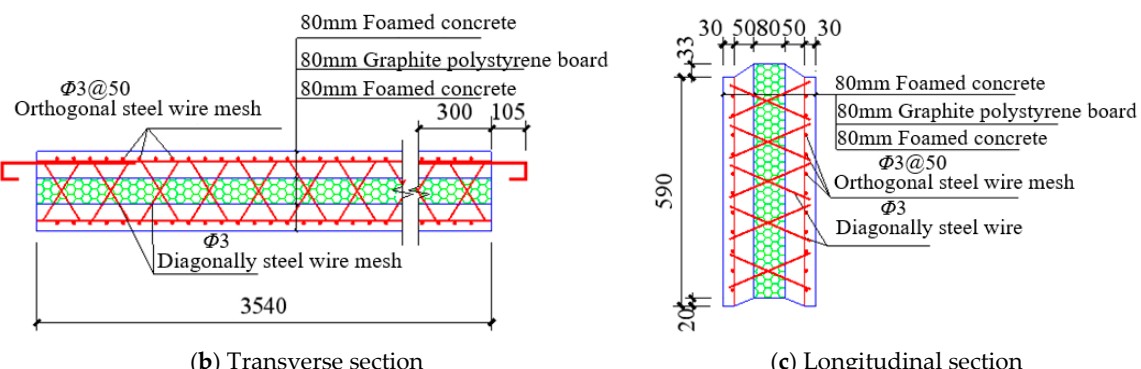

(**b**) Transverse section         (**c**) Longitudinal section

**Figure 2.** Structural design of the composite panel.

A wall with an opening was formed by splicing sandwich panels both with and without openings. Panels with openings were designed with a steel tie bar at the edge for connection with the framework with no steel tie bar at the opening edge. Either a door or a window was connected to the edge without a steel tie bar. The length of a panel with an opening can be changed according to different sizes of doors and windows.

The door opening of the specimen measured 1800 mm in height and 800 mm in width. The panels for the door opening measured 1370 mm in length. The window opening of the specimen measured 1000 mm in height and 1400 mm in width. The panels for the window opening measured 1070 mm in length.

The tongue-and-groove joint between the two sandwich panels had the tongue along the top edge and the groove along the bottom edge of the panels. By inserting the groove along the bottom edge of one panel into the tongue along the top edge of a panel, the perpendicularity of the two joined panels was ensured. The seams between the joined panels were filled with foamed concrete and then scraped until it was level. A slab formed by joining the sandwich panels functions as an integrated

whole because of the bonding enabled by the tongue-groove mechanism and the foamed concrete. Figure 3 showed the mechanism and seam of the joint between the two panels.

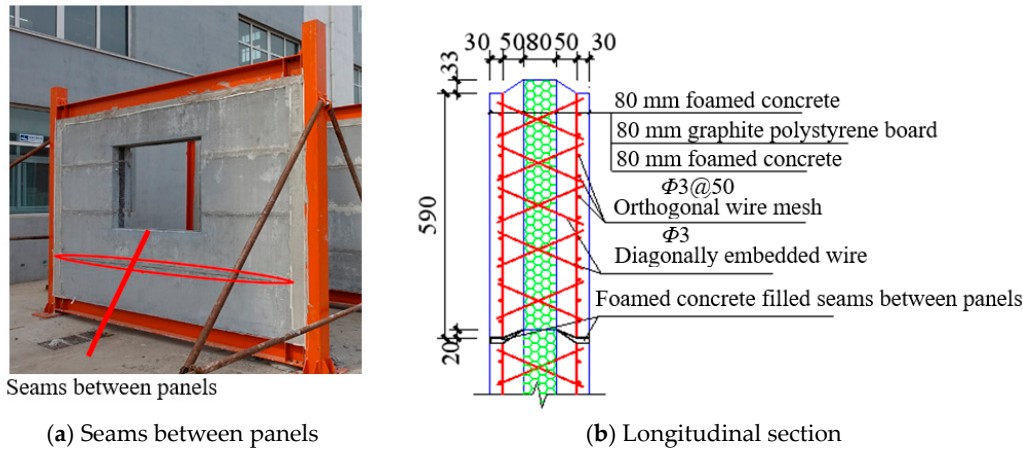

| (**a**) Seams between panels | (**b**) Longitudinal section |

**Figure 3.** Mechanism and seam of the joint between two panels.

A concealed steel-truss panel comprised of a lightweight steel truss and foamed concrete was built similarly to the element of a short-pier shear wall for resisting lateral forces. Because a large opening of a wall structure results in a large reduction of its lateral force resistance, concealed steel-truss panels were placed at the two sides of the opening.

The chords of the lightweight steel-truss were made from a square steel tube ($100 \times 100 \times 6$ mm) filled with recycled concrete C40. The web members of the lightweight steel truss were made from a round steel tube measuring 50 mm in diameter and 4 mm in wall thickness. It was assembled with a horizontal inclination of $45°$. The chords and web members were subsequently welded together. Foamed concrete was cast around the lightweight steel-truss to completely conceal it in the foamed concrete. The openings between piers could be used as openings for bay windows and other windows but were retained throughout the experiment. Figure 4 showed a concealed steel-truss panel.

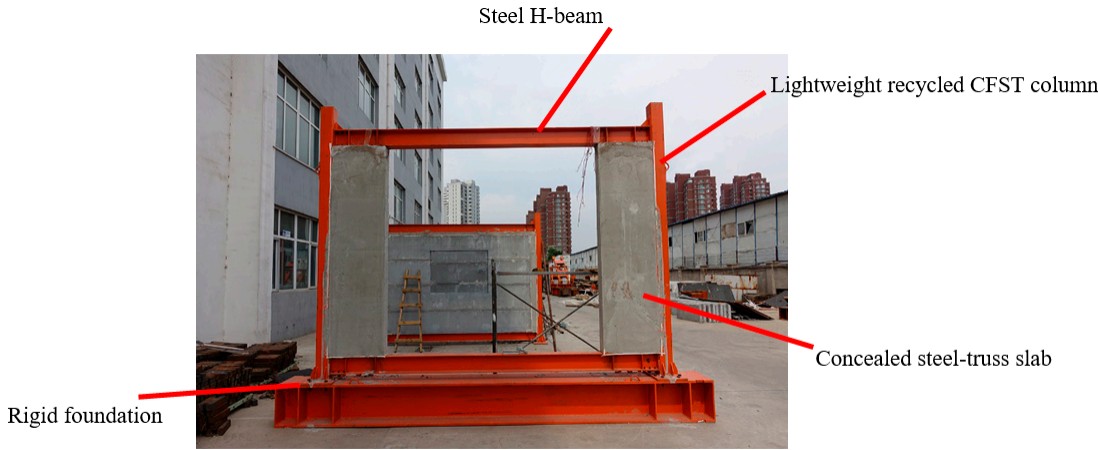

**Figure 4.** Concealed steel-truss panel.

### 2.1.3. Design of the Framework–Panel Connection

The connection between a framework and a sandwich panel was realized by the steel-tie bars attached to the panel. The exposed part of the steel-tie bar was extended horizontally from within the panel and was then bent at an angle of $90°$. The length of the horizontal section was 105 mm and the length of the bent section was 40 mm. A sandwich panel was assembled to a framework by continuously welding the bent section of the exposed steel tie bar to the framework columns,

as shown in Figure 5a. A gap was reserved between the sandwich panel and the CFST column and filled with back-cast polyphenylene particle foamed concrete, as shown in Figure 5b,c with details of the panel–column connection.

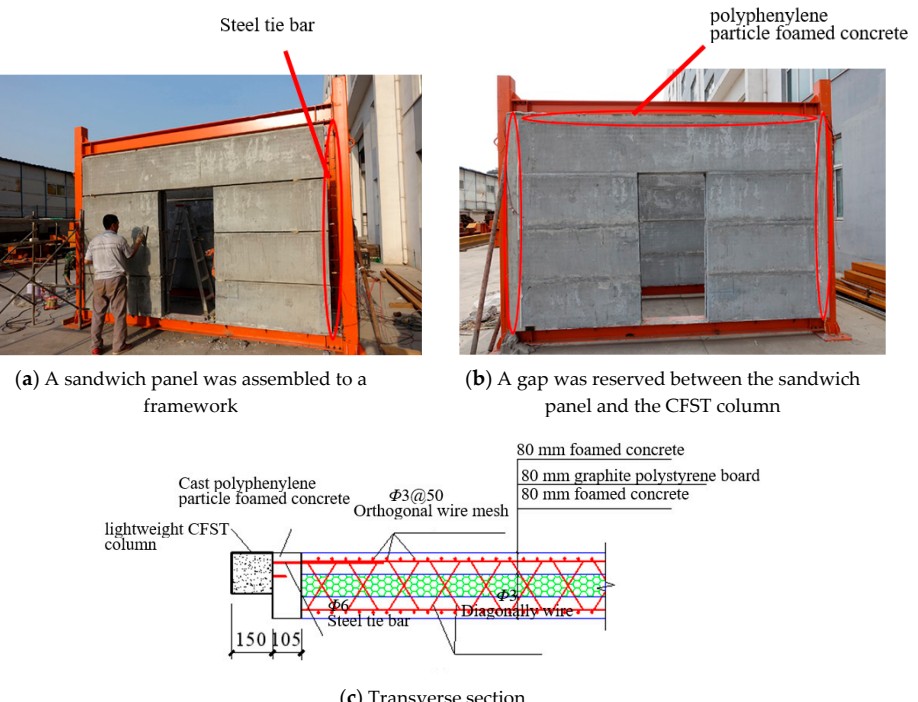

(**a**) A sandwich panel was assembled to a framework

(**b**) A gap was reserved between the sandwich panel and the CFST column

(**c**) Transverse section

**Figure 5.** Details of the panel connection.

The framework column was connected to a concealed steel-truss panel by bolting and welding. A connecting plate was welded to each end of a chord (i.e., concealed lightweight recycled CFST column) of the lightweight truss, with the width of the connecting plate the same as the width of the flange of the steel H-beam. The connecting plate was connected with the flange of the steel H-beam using high-strength bolts. A web member (i.e., round steel tube) of the lightweight truss was welded to the chord of the truss at one end and welded to a framework column (i.e., lightweight CFST column) at the other. The truss and the framework were in the same plane. Polyphenylene particle foamed concrete was then cast around the truss and bound with the framework. Figure 6a showed the connection between the truss chord and the steel H-beam. Figure 6b showed the connection between the web member of truss and the framework column.

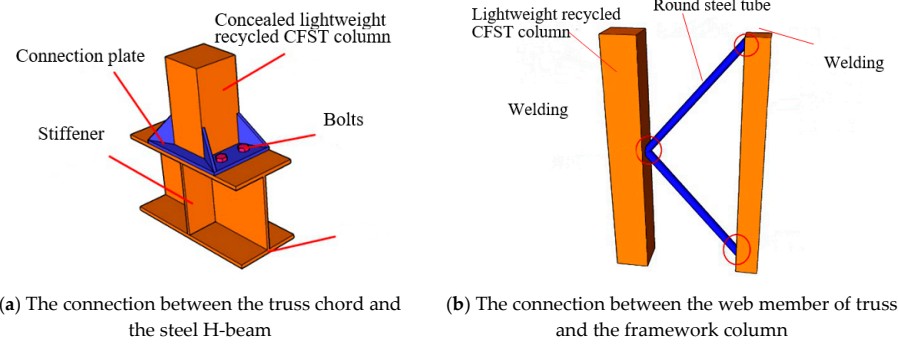

(**a**) The connection between the truss chord and the steel H-beam

(**b**) The connection between the web member of truss and the framework column

**Figure 6.** Connecting the chord of the truss and the steel H-beam and the web member of the truss.

2.1.4. Assembly of the Framework and the Slab Structure

The framework and slab structure required only a very few assembly workers. Therefore, it facilitated rapid and high-quality construction. In the following section, the procedure and method for assembling the framework-slab structure were illustrated for the case of the sandwich-panel walled framework without an opening. The assembly process was divided into the following three stages including framework assembly, panel assembly, and back-casting to fill the framework-panel gap. Framework assembly fixed the columns perpendicularly onto the foundation beam, lifted the beam of the framework, and moved it horizontally into the double stiffener-reinforced brackets to align the mounting holes in the brackets with those of the beam flange. The beam and brackets were connected with M12 high-strength bolts, as shown in Figure 7a. Assembly of sandwich panels placed a panel laterally into the framework, as shown in Figure 7b. A second panel was placed atop the first to ensure the groove along the bottom edge of the first was properly inserted into the tongue along the top edge of the second panel. The seam was then filled with foamed concrete, as shown in Figure 7c. When all sandwich panels were placed, the exposed steel tie bars were welded to the slab of framework columns, as shown in Figure 7d. Back-casting filled the gaps between the sandwich panels and framework columns by back-casting polyphenylene particle foamed concrete, as shown in Figure 7e. The mould was removed after the concrete cures, as shown in Figure 7f.

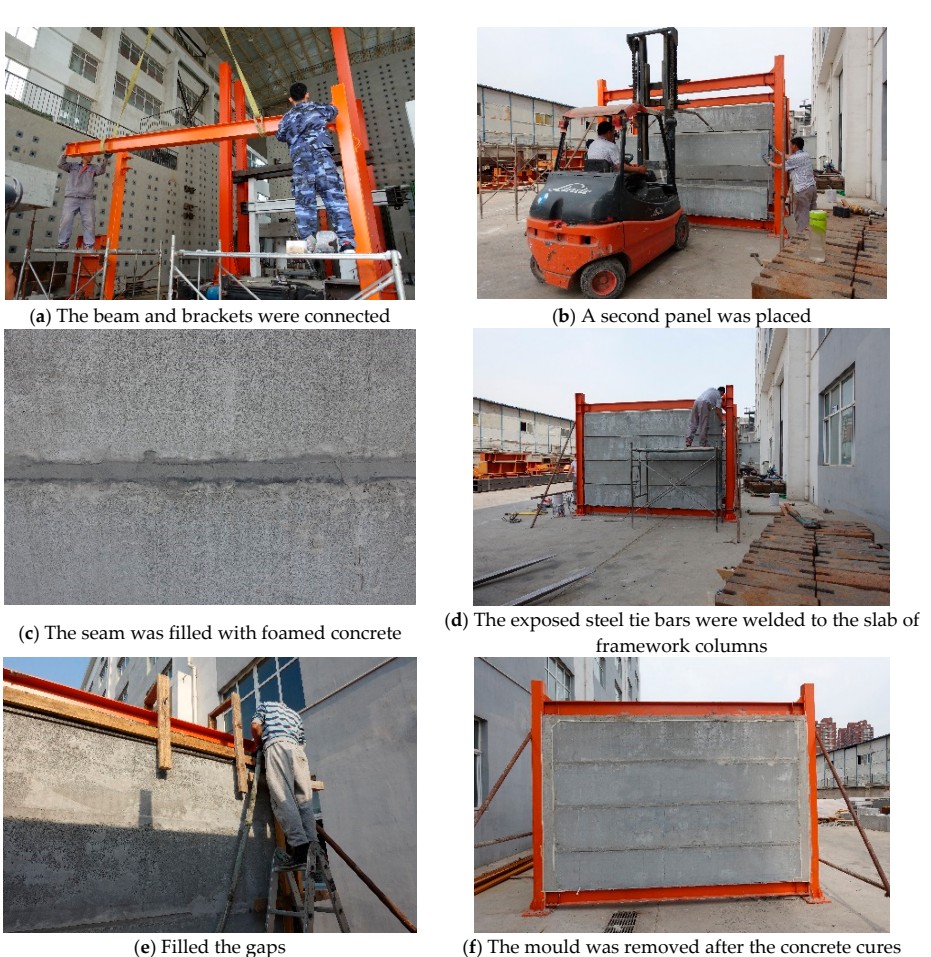

(**a**) The beam and brackets were connected

(**b**) A second panel was placed

(**c**) The seam was filled with foamed concrete

(**d**) The exposed steel tie bars were welded to the slab of framework columns

(**e**) Filled the gaps

(**f**) The mould was removed after the concrete cures

**Figure 7.** Framework-slab assembly.

Table 1 showed the numbering and parameters of the prepared specimens. Figure 8 showed the major dimensions of the specimens.

**Table 1.** Numbering and parameters of the prepared specimens.

| Specimen No. | Specification of Framework Column | Specification of Framework Beam | Specification of Slab | Type of Opening | Size of Opening (mm) |
|---|---|---|---|---|---|
| FM | | | - | - | - |
| FM-S | lightweight recycled CFST column | Steel H-beam | Sandwich slab without opening | - | - |
| FM-SW | | | Sandwich slab with a window opening | Window opening | 1400 × 1000 |
| FM-SD | | | Sandwich slab with a door opening | Door opening | 800 × 1800 |
| FM-SS | | | Concealed steel-truss slab | Window opening | 2480 × 2436 |

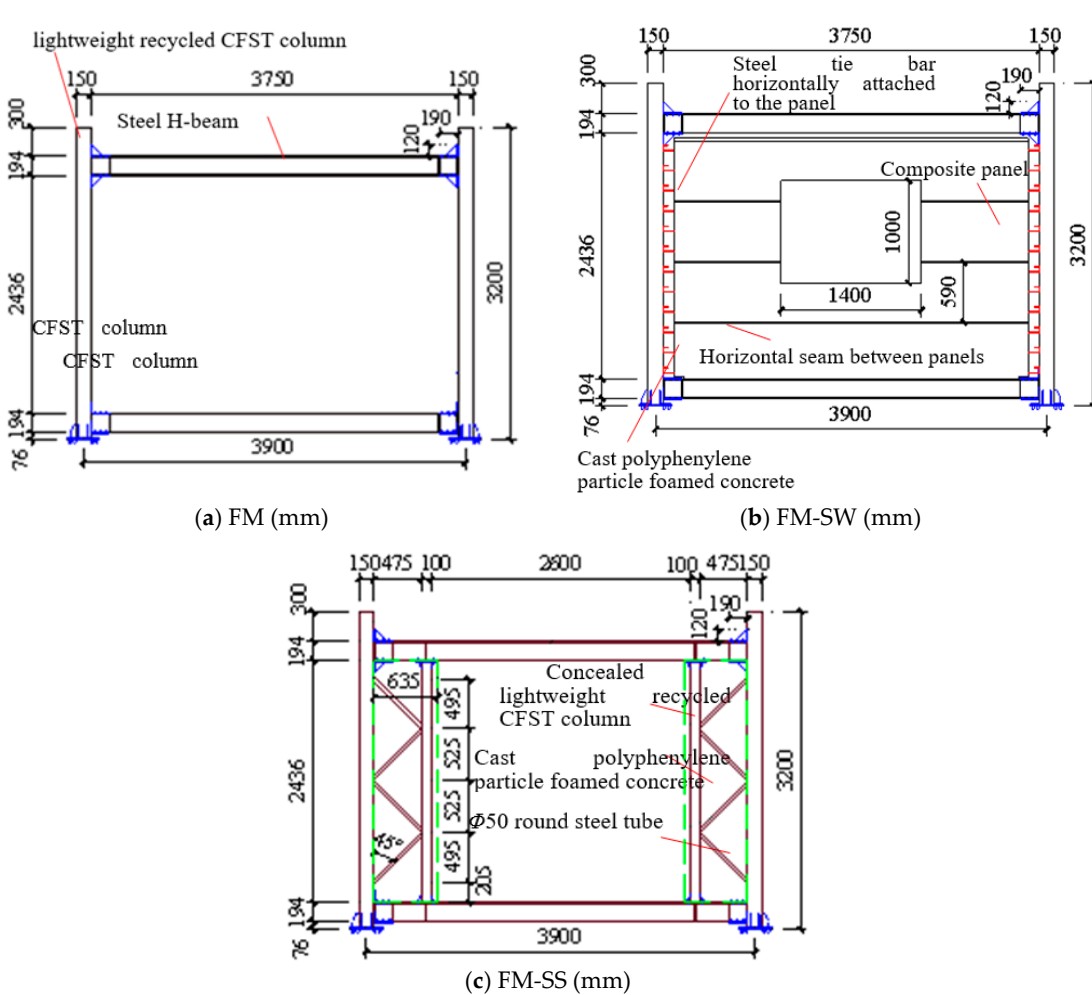

(**a**) FM (mm)          (**b**) FM-SW (mm)

(**c**) FM-SS (mm)

**Figure 8.** Major dimensions of the specimens illustrated with specimens FM, FM-SW, and FM-SS.

## 2.2. Material Property Test

### 2.2.1. Properties of Steel Materials

According to the requirements of Steel and steel products–location and preparation of samples and test pieces for mechanical testing (GB/T2975–1998)' [25], samples were obtained from the corresponding locations of the test members. Three standard tensile specimens were conducted for each steel type, according to the requirements of 'Metallic materials—tensile testing—Part 1: Method of test at room temperature (GB/T228.1–2010)' [26]. The specification of the cold-formed square steel tube, round seamless steel tube, hot-rolled steel H-beam, and fabricated steel connection parts was Q235b. Table 2 showed the mechanical properties of steel bars and plates.

**Table 2.** Mechanical properties of steel bars and plates.

| Steel Product | Sampling Location | Diameter/Thickness $t$ (mm) | Yield Strength $f_y$ (MPa) | Ultimate Strength $f_u$ (MPa) | Modulus of Elasticity $E$ (GPa) | Elongation $\delta$ (%) |
|---|---|---|---|---|---|---|
| Steel-tie bar | Sandwich panel | 6 | 405.0 | 581.0 | 206.9 | 16.1 |
| Galvanised cold-drawn wire | Sandwich panel | 3 | 662.0 | 718.0 | 190.8 | 3.1 |
| Square steel-tube wall | lightweight CFST column | 6 | 373.0 | 444.3 | 218.2 | 21.5 |
| Flange of steel beam | Steel H-beam | 9 | 282.7 | 431.0 | 195.1 | 16.1 |
| Web of steel beam | Steel H-beam | 6 | 296.0 | 453.0 | 202.2 | 30.7 |
| Connection L-bracket | Beam-column connection | 8 | 318.0 | 468.0 | 202.9 | 19.1 |
| Square steel-tube wall | Chord of steel truss | 6 | 414.0 | 552.0 | 204.8 | 27.3 |
| Round steel-tube wall | Web of steel truss | 4 | 366.3 | 424.3 | 231.2 | 18.6 |

### 2.2.2. Properties of Concrete Materials

The lightweight steel tube recycled concrete column was filled with C40 recycled concrete with a coarse aggregate replacement rate of 100%. Recycled coarse aggregate was processed and produced by the Shougang Resources Science and Technology Development Company in Beijing (Figure 9). The physical properties of the coarse aggregate are listed in Table 3.

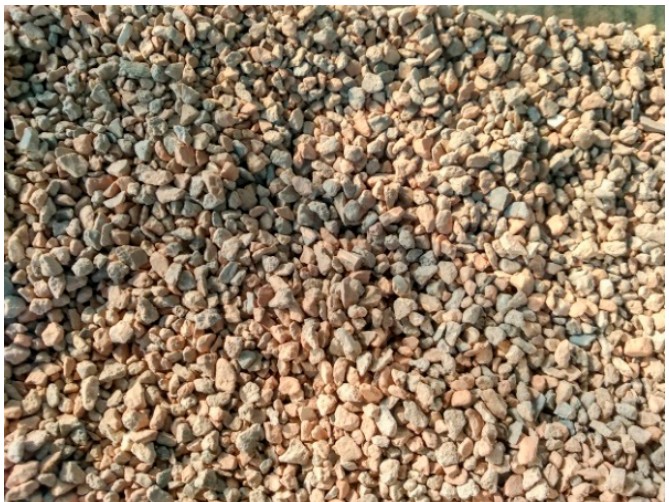

**Figure 9.** Recycled coarse aggregate (5 to 10 mm).

**Table 3.** Physical properties of the recycled coarse aggregate.

| Grain Size (mm) | Apparent Density (kg/m³) | Water Absorption (%) | Crushing Index (%) | Void Content (%) | Content of Elongated and Flaky Particles (%) |
|---|---|---|---|---|---|
| 5–10 | 2650 | 4.45 | 9.0 | 48.0 | 4.0 |

Fine sand was added as a fine aggregate when mixing the recycled concrete. The water content of the coarse aggregate on the day of casting was 3.92% and that of the fine aggregate was 6.02%. Table 4 showed the mix proportions of the recycled concrete.

**Table 4.** Mix proportions of the recycled concrete.

| Design Strength | Mix Proportion (kg/m$^3$) | | | | | | |
|---|---|---|---|---|---|---|---|
| | 42.5 Cement | Fly Ash | Mineral Powder | Recycled Coarse Aggregate | Fine Sand | Water Reducing Agent | Water |
| C40 | 323.0 | 70.0 | 70.0 | 804.0 | 825.0 | 4.3 | 16.5 |

Three test cubes (150 × 150 × 150 mm) and three test prisms (150 × 150 × 300 mm) were prepared for each batch of recycled concrete and were cured under the same conditions as the specimens. Table 5 showed the material properties as tested.

**Table 5.** Material properties.

| Compressive Strength of Test Cube on the 28th day $f_{cu, m}$ (MPa) | Axial Compressive Strength on the 28th day $f_{c, m}$ (MPa) | Modulus of Elasticity $E_c$ (GPa) |
|---|---|---|
| 54.0 | 35.5 | 30.2 |

The compressive strength of the test cube of the foamed concrete for casting sandwich panels was 3.8 MPa, as tested on the 28th day. The compressive strength of the test cube of the polyphenylene particle-foamed concrete, mixed with polyphenylene particles for filling the gap between the sandwich panels and framework, was 1.9 MPa, as tested on the 28th day. The same batch of polyphenylene particle foamed concrete was used to cast the concealed steel-truss panels.

The sites where the material property tests were conducted and shown in Figure 10.

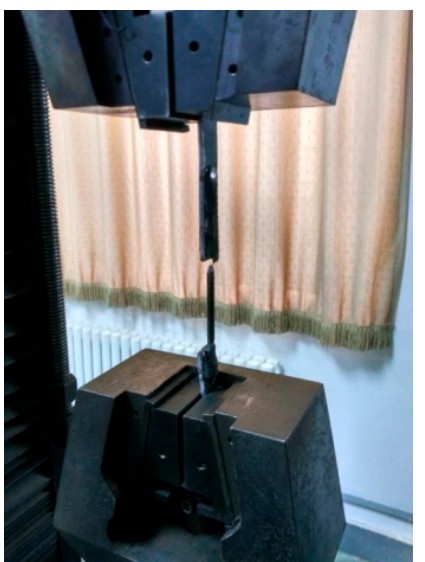 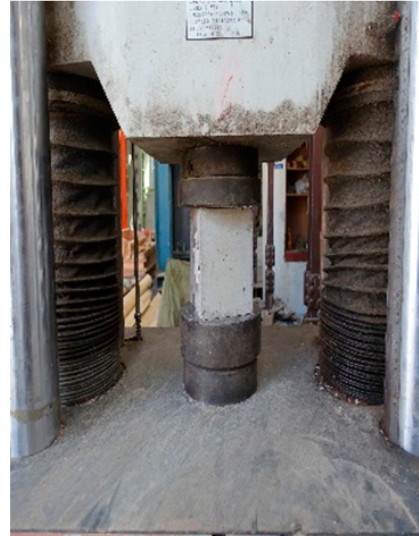

(**a**) Material property test for steel    (**b**) Material property test for concrete

**Figure 10.** Material property tests.

*2.3. Experimental Scheme*

The specimens were tested under displacement-based loading with the loading determined by the displacement angle, θ. More specifically, the loading was increased by an increment of 0.125% when θ ≤ 0.5%, by 0.25% when 0.5% < θ ≤ 2%, and by 0.5% when θ > 2%. Each increment in loading was repeated twice. Figure 11 showed the loading protocol.

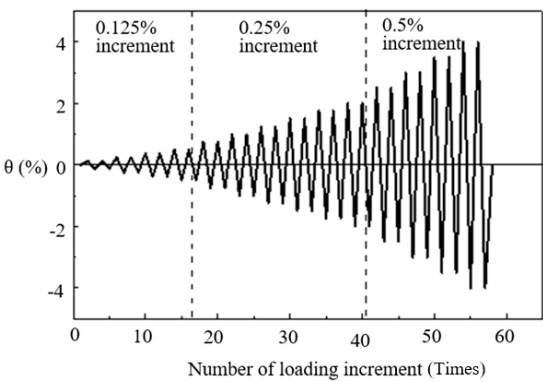

**Figure 11.** Results of loading protocol.

During the test, a vertical load was applied at the top of each framework column, and a cyclic reversed lateral load was applied along the axis of the framework beam. The vertical load was provided by two 250-T jacks. The axial compression ratio adopted for the experiment was 0.28, and the load computed, according to this ratio was 499.5 kN. The vertical load applied to the top of the framework columns was increased to the pre-established value in a stepped manner to ensure a consistent loading atop of the columns. The structural system of the study was mainly suitable for low-rise (one–three story) residential houses and the axial compression ratio of the columns was determined, according to the larger axial pressure of the columns in the three-story residential structure). A lateral load was applied in a slow and continuous manner, at a loading rate of 0.5 mm/s. The testing of a specimen was ended when a shear failure occurred with the bolts, a clear bending occurred with the L-brackets, or the lateral load decreased to below 85% of the ultimate load. Figure 12 showed the configuration of the loading devices. Figure 13 showed the arrangement of the measurement points.

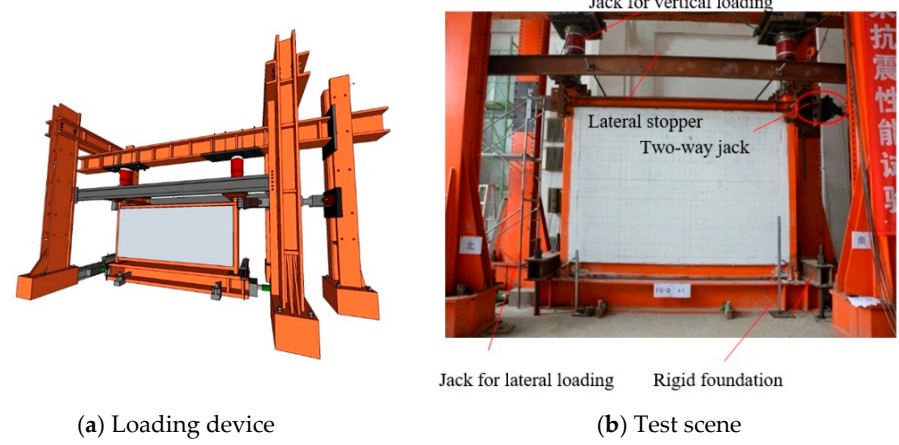

(**a**) Loading device          (**b**) Test scene

**Figure 12.** Configuration of the loading devices.

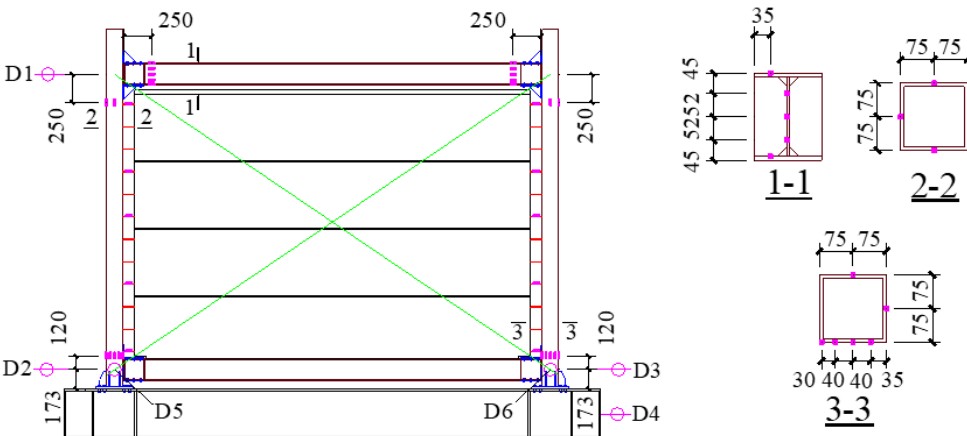

**Figure 13.** Arrangement of the measurement points.

### 2.4. Damaging Process

Specimens FM-S, FM-SW, FM-SD, and FM-SS exhibited a similar failure development mechanism. The failure started in the slabs and expanded to the framework. More specifically, horizontal cracks developed along the seams between the sandwich panels, sliding between the panels, as shown in Figure 14a. Cracks then developed in the foamed concrete-filled seams and extended along this structure, which indicates that the lateral load applied to the sandwich panels was greater than the sum of the friction and cohesion at the seams. Consequently, a slab specimen disintegrated into individual sandwich panels. Then, a bending deformation developed in the framework columns resulted in inconsistent deformation between the sandwich panels and the framework. This inconsistent deformation, in turn, developed a diagonal compression from the framework onto the sandwich panels, and, consequently, to lateral sliding in the corners of the panels. Oblique cracks were found in the corners of the slabs, and, as the loading increased, they exhibited the phenomenon of repeated opening and closing, as shown in Figure 14b.

The many oblique cracks in the corners of the slabs could be explained by the following mechanism. When a specimen was subjected to an increasing load, the lateral load was transmitted to the sandwich panels via the steel-tie bars welded to the framework columns. This caused the steel tie bars to be subject to repeated tensile and compressive stresses, which resulted in higher stresses in the corners. Because the slabs and framework had different deformabilities, wider oblique cracks developed along the interfaces of the polyphenylene particle foamed concrete cast to fill the gap between the panels and the framework. When the loading on a specimen increased to a displacement angle of 4% (Δ = 111.32 mm), the lower part of the framework column bent outward slightly, as shown in Figure 14c. The testing of the specimen was terminated when this occurred. Then, clear oblique cracks developed around the corners of the openings of specimens FM-SW and FM-SD and the cast-foamed concrete was crushed, as shown in Figures 12e and 14d. This could be explained by stress concentrations occurring around the corners of the openings.

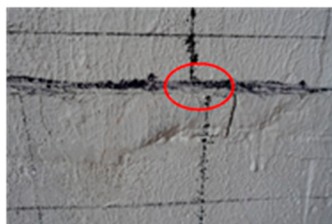

(**a**) Horizonal sliding between composite panels. Distance of sliding reached 5mm

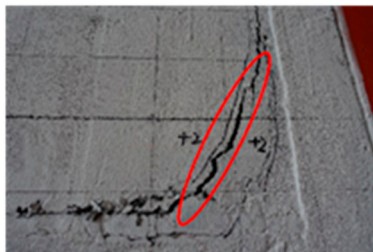

(**b**) Oblique cracks in the corners of the slab. The width of the biggest crack was 10 mm approximately

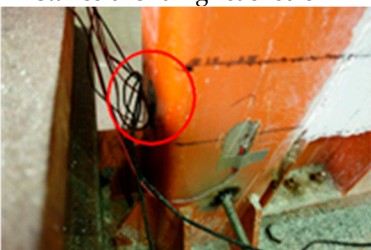

(**c**) Slight outward bending in the lower part of the column

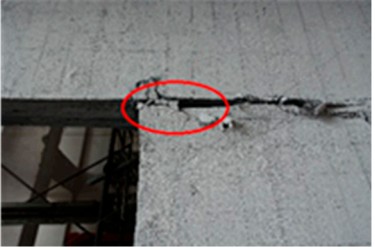

(**d**) Crushed concrete around the corner of the window opening

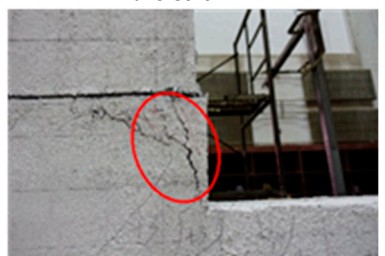

(**e**) Crushed concrete around the corner of the door opening

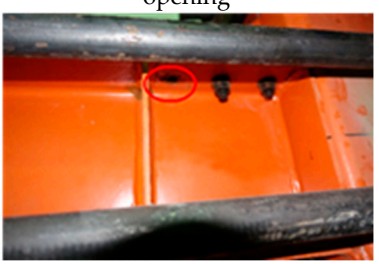

(**f**) Sheared-off bolts for joining the beam and columns of specimen FM

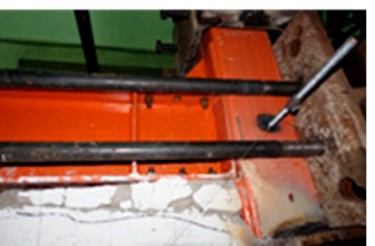

(**g**) Sheared-off bolts for joining the beam and columns of specimen FM-SS

**Figure 14.** Experimental phenomena and failure of the specimens.

Regarding the specimen FM, a bending deformation developed in the overall framework as the loading increased. Additionally, the joints of the framework were sheared off at a higher lateral load, as shown in Figure 14f. The test was terminated when this failure occurred. This result could be explained by the fact that the lateral load on the specimen was born completely by the beam of the framework, whereas the slabs mounted in the framework shared the lateral load on the other specimens.

Regarding specimen FM-SS, the locations of the cracks that developed in the concealed steel-truss panel basically matched the locations of the chords and web members of the truss. This was caused by the smaller deformation of the chords and web members of the truss and the greater deformation of the polyphenylene particle foamed concrete cast around the truss. It was this inconsistent deformation that resulted in cracks developing in the concrete over the truss members. Furthermore, bending of the flange of the steel beam occurred at the joints of the lightweight truss and the frame beam, or at the foundation beam of the framework. Due to the greater overall stiffness of the concealed steel truss panel, the slab of this specimen was subjected to a greater lateral load. The loading of the specimen was

transmitted via the joints between the lightweight steel-truss and the frame beam or foundation-beam of the framework and the steel structural parts of the joints that were prone to bending deformation.

The framework of specimen FM-SS was the same as those of specimens FM, FM-S, FM-SW, and FM-SD. However, no outward bending developed in the lower part of the columns of specimen FM-SS. In the case of this specimen, the lightweight steel truss acted with the framework as a complete load-bearing unit. Consequently, it supported the majority of the lateral load acting on the framework columns. The framework of specimen FM-SS was considered failed when the bolts for joining the beam and columns of the framework were sheared off, as shown in Figure 14g.

The double L-shaped joint with the stiffener rib of all specimens exhibited no major deformation throughout the test (Figure 15). Upon the failure of the specimens, the beams and columns of the joint remained perpendicular despite the reinforced state of the joint. This would lead to increased stiffness, which could be explained by the addition of stiffening ribs to the L-shaped joint. The design improved the strength and the stiffness of the joint areas since the composite structure complied with the model of weak members and strong joints.

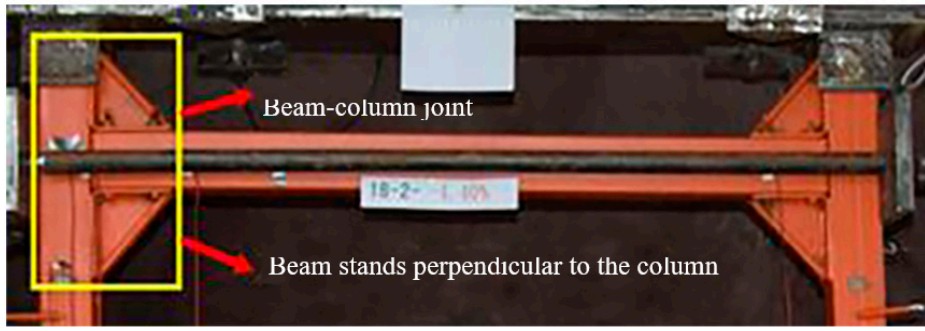

**Figure 15.** Demonstration of occurrence of damage to beam–column joints.

## 3. Results and Analysis of Prefabricated Lightweight CFST Framework-Composite Slab Structure

*3.1. Hysteretic Performance*

Figure 16 showed the load (*F*)-displacement (Δ) curves of the tested specimens.

Figure 16 showed that the hysteresis curve of specimen FM was clearly different from the hysteresis curves of the specimens with sandwich panels. The hysteresis curve of specimen FM did not exhibit a turning point as the load increased, which indicates smaller load-bearing and a smaller-energy dissipation capacity. The curve was shuttle-shaped and did not exhibit a clear centralization trend.

Regarding specimens FM-S, FM-SW, and FM-SD, the slopes of the hysteresis curves increased with an increase in displacement and load, which exhibits a turning point right before the displacement reached the target value. It then decreased with an increase in displacement. The curves under the same direction of loading show that the slope under a later loading increment was clearly smaller than that under an earlier increment. Additionally, after repeated loading increments, the hysteresis curves of the specimens gradually became more full-rounded, which changes from a reverse-'S' shape to a 'Z' shape.

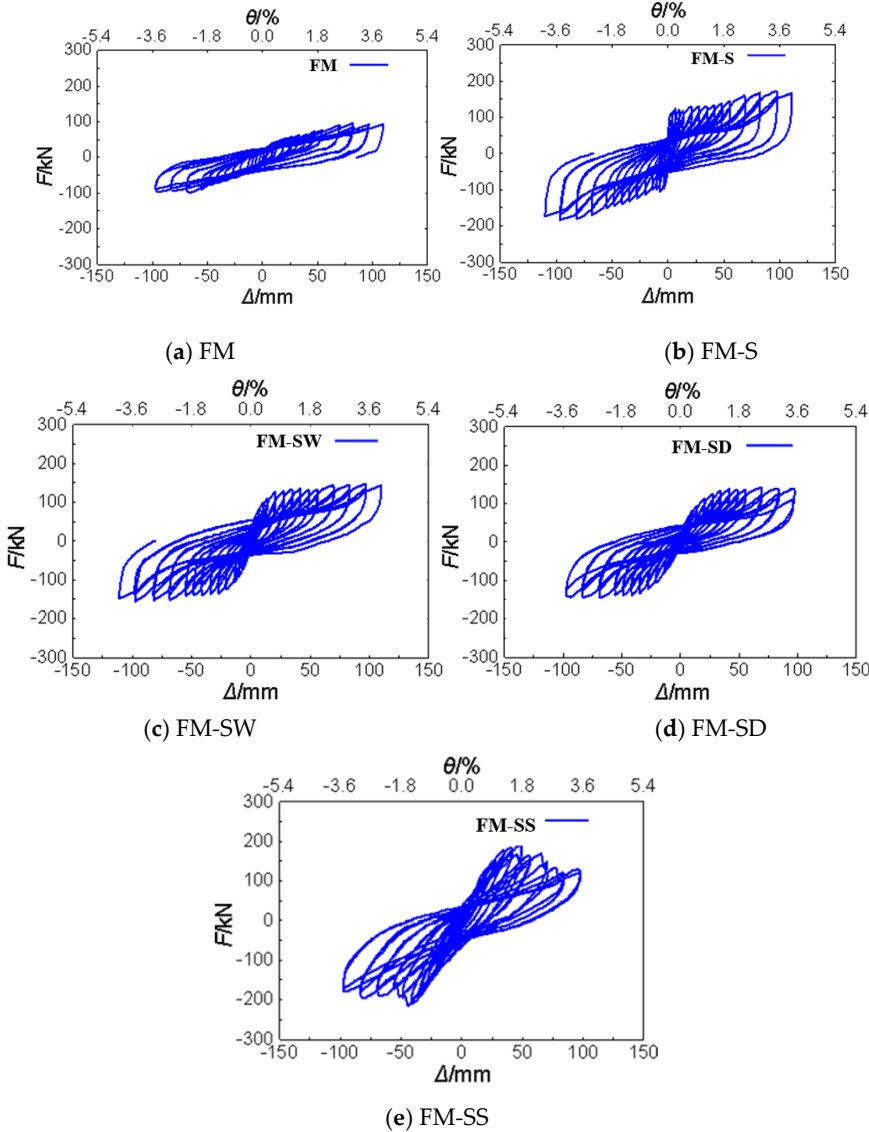

**Figure 16.** Hysteresis curves of the tested specimens.

When the sandwich slabs and framework underwent synergistic deformation, the slabs continuously squeezed the framework, serving as a support structure. Consequently, the slope of the hysteresis curve increased with the load. When the slabs cracked, and plastic deformation occurred in the framework, the slope of the curve decreased with the load increase. The phenomenon where the slope of the hysteresis curve decreased with an increase in the number of load increment indicates increasing damage to the framework and the slabs. As the load increased, greater sliding between the sandwich panels occurred, and the friction between the sandwich panels and the plastic deformation of the steel tubes of the framework resulted in energy dissipation. Consequently, the hysteresis curve became more fully rounded.

The hysteresis curves of specimen FM-SS was the reverse-'S' shape, which has a smaller enveloped area and indicates smaller energy-dissipation capacity of the structure. However, the specimen exhibited a higher load-bearing capacity at displacement angles smaller than 1.8%. A comparison of the curves under the same direction of loading revealed greater slopes and smaller reductions of the hysteresis curves prior to the point of the ultimate displacement angle (i.e., the displacement angle corresponding to the ultimate load), and a drastic decrease of the slopes of the hysteresis curves beyond the point of the ultimate displacement angle.

Additionally, the authors had carried out a shaking table test on a full-scale and two-story prefabricated lightweight concrete-filled steel tubular (CFST) framework composite slab structure, with a building height of 5400 mm, and plane size dimensions of 4400 × 4400 mm. The experimental results showed that the structure was subjected to an earthquake of 8°. The inter-story displacement angle was approximately equal to 1/500. There was no clear damage to the structure. This showed that the structure had good seismic performance. Accordingly, the structure exhibited functional recoverability at large earthquake intensities. Although the shape of the hysteretic curve was not full, the functionality could be recovered after the earthquake.

### 3.2. Analysis of Skeleton Curves

Figure 17 showed the skeleton curves of the tested specimens.

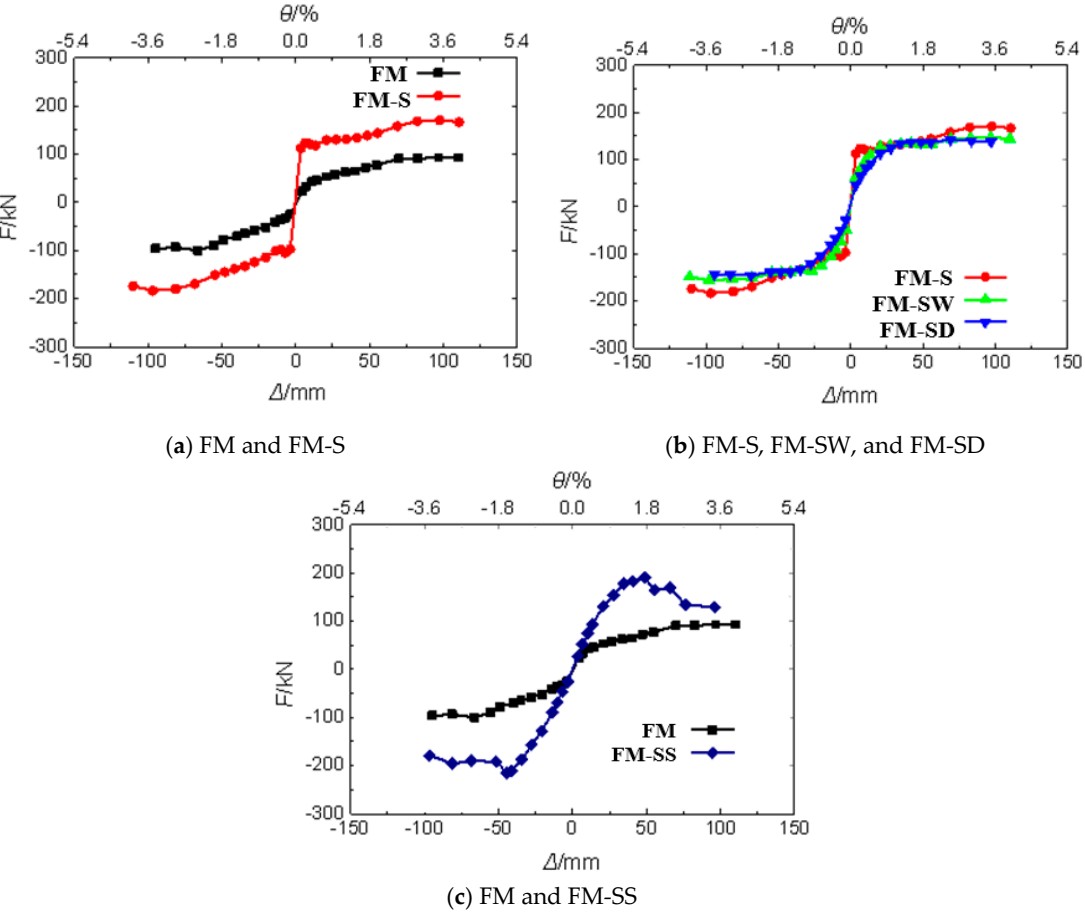

**Figure 17.** Skeleton curves of the tested specimens.

In Figure 17a, the framework skeleton curve of specimen FM could be fitted or simplified as a four-section broken line. The first section represented the linear elastic stage. The second section represented an approximately linear stage from the point when the framework started to yield to the point when the framework exhibited clear yielding. The third section represented a linear stage from a clear yield to the limit of the load increase. The fourth section was a flat, straight line that represents the stage from the point of the load limit to a decreasing load on the specimen. The fourth stage of specimen FM corresponded to the fourth stage of specimen FM-S, which were both represented as a flat, straight line.

Figure 17b showed that, when sliding between the sandwich panels occurred and cracks developed along the seams between them, specimens FM-SW and FM-SD deteriorated more rapidly in load-bearing capacity than specimen FM-S. This process continued approximately until the end

of the flat, straight line representing the second stage of the framework curve of specimen FM-S. Compared with specimen FM-S, the contribution of the sandwich panels to the load-bearing capacities of specimens FM-SW and FM-SD decreased more rapidly in the earlier stages but remained more stable in later stages. Additionally, the contribution of the sandwich panels to the load-bearing capacity did not exhibit a clear decrease even after the loads on specimens FM-SW and FM-SD reached a limit. Regarding specimens FM-SW, FM-SD, and FM-S, clear sliding and cracking developed along the seams between the sandwich panels. Additionally, the ultimate loads on specimens FM-SW and FM-SD were smaller due to the effects of the door and window openings, respectively.

Figure 17c showed that the skeleton curve of specimen FM-SS could be simplified as a three-section broken line. The first section represented a linear elastic stage, which ended at a displacement angle of approximately 0.75°. The second section represented a stage from the structural yield point through the limit of the load increase. The third section represented the stage of a gradual decrease in structural load-bearing capacity.

Specimen FM-SS reached its ultimate load at a displacement angle of approximately 1.8°. At lower displacement angles, specimen FM-SS exhibited a clearly higher load-bearing capacity than specimen FM, which indicated that the concealed steel-truss slab contributed to a significantly higher load-bearing capacity of the lightweight CFST framework. Prior to the yielding of the framework, specimen FM-SS exhibited a higher lateral displacement-resistance stiffness than specimen FM. After the framework yielded, specimen FM-SS deteriorated more slowly in stiffness than specimen FM. This indicated that the concealed steel-truss slab contributed to a higher lateral displacement-resistance stiffness of the lightweight CFST framework and, thus, effectively confined the deformation of the framework. Additionally, in the decreasing load stage, the load borne by specimen FM-SS decreased more slowly, which indicated a better structural ductility.

## 3.3. Characteristic Points of the Skeleton Curves

For the experiments, the characteristic points were selected in accordance with several rules. The cracking status referred to the loading and displacement observed as minor cracks started to appear on the slab. The ultimate status and failure status were determined based on collected data, whereas the yielding status was calculated using the energy equivalence method. The frameworks of specimens did not yield when the specimens yielded, except for specimen FM. Considering this fact, the yield of the frameworks was defined as the status when the strain in the lower part of framework columns reached the yield strain. Table 6 showed the loads and displacement angles corresponding to the characteristic points of the skeleton curves of the tested specimens.

**Table 6.** Characteristic points of the skeleton curves of the tested specimens.

| Specimen No. | Yield | | Ultimate | | Failure | | Yield of the Framework | |
|---|---|---|---|---|---|---|---|---|
| | $F_y$ (kN) | $\theta_y$ (%) | $F_{max}$ (kN) | $\theta_{max}$ (%) | $F_u$ (kN) | $\theta_u$ (%) | $F_{fy}$ (kN) | $\theta_{fy}$ (%) |
| FM | 64.20 | 1.23 | 98.09 | 2.68 | 94.68 | 3.68 | 62.39 | 1.23 |
| FM-S | 110.42 | 0.35 | 176.83 | 3.49 | 170.50 | 3.96 | 126.78 | 0.88 |
| FM-SW | 114.43 | 0.59 | 151.75 | 3.50 | 146.25 | 3.98 | 132.75 | 0.85 |
| FM-SD | 113.94 | 0.82 | 144.08 | 2.48 | 140.68 | 3.44 | 116.45 | 0.87 |
| FM-SS | 186.27 | 1.28 | 202.90 | 1.68 | 153.78 | 3.47 | 155.51 | 1.00 |

Displacement angle $\theta$ (%)—displacement at feature point ($\triangle$/mm)/distance from the embedded end of the frame column to the loading point (h/mm) $\times$ 100%.

An analysis of Table 6 revealed that the yield load, ultimate load, and yield load of the framework of specimen FM-S were 1.72, 1.80, and 2.03 times those of specimen FM, respectively. The ultimate loads of specimens FM-SW and FM-SD were 85.2% and 79.6% of the ultimate load of specimen FM-S, respectively. The yield loads of specimens FM-SW and FM-SD were 103.6% and 103.2% of the yield load of specimen FM-S, respectively, which indicates that an opening in the sandwich slab

contributed to a smaller ultimate load of the specimen, but did not have a clear effect on the yield load of the specimen.

The area of the window opening along the edges of the sandwich slab of specimen FM-SW was 2.86% smaller than the door opening where the sandwich slab was broken at the bottom of specimen FM-SD. However, at a displacement angle of 1%, the load borne by the first specimen was 8.27% greater than that of the second specimen. Moreover, the ultimate load of the first specimen was 5.32% greater than that of the second specimen, which indicates that the type of opening in the sandwich slab had some degree of impact on the ultimate load of the structure, with a greater impact of the door opening on the ultimate load.

The frameworks of specimens FM-S, FM-SW, and FM-SD yielded a similar displacement angle, which indicates that an opening in the sandwich slab had no impact on the yield deformation of the framework. The yield load, ultimate load, and yield load of the framework of specimen FM-SS were 2.90, 2.07, and 2.68 times those of specimen FM, respectively, which indicates that the concealed steel-truss slab contributed to a much higher load-bearing capacity of the lightweight CFST framework.

Although the sandwich wall was formed by connecting multiple sandwich panels with tongue-and-groove joints, the working performance of the wall structure was similar to a seamless wall under loads smaller than the yield load. Under loads greater than the yield load, clear sliding occurred between the sandwich panels along the seams between them and the articulation and friction between the panels along the tongue-and-groove joints still contributed to the overall strength of the wall. The deteriorated working performance of the wall when subjected to loads greater than the yield load was considered by introducing a seam effect coefficient.

Since the support provided by a panel with a window or door opening was smaller than that provided by a panel without opening, and the support provided by a wall panel a door opening was smaller than the support provided by a panel with a window opening of the same opening ratio, a reduction coefficient for the type of opening and for the opening ratio could be introduced to account for these effects [12,27]. It was given by Equation (1) shown below.

$$\alpha = \lambda \left(1 - A / h_{inf} \cdot l_{inf}\right) \tag{1}$$

where A was the area of the opening, $h_{inf}$ was the height of the sandwich wall of the specimen, $l_{inf}$ was the width of the sandwich wall of the specimen, and $\lambda$ was the coefficient for the type of opening (0.8 for a door opening, 0.9 for a window opening, and 1.0 for no opening).

### 3.4. Deterioration in Strength and Stiffness

The strength reduction factor, $\eta$, was the ratio of the ultimate load in the last cycle of loading increment to a target displacement, to the ultimate load of the first cycle of loading increment to the same target displacement. This factor was used to reflect the deterioration in the load-bearing capacity of a specimen. Figure 18 showed variations in the strength deterioration factor with the displacement angles of the specimens.

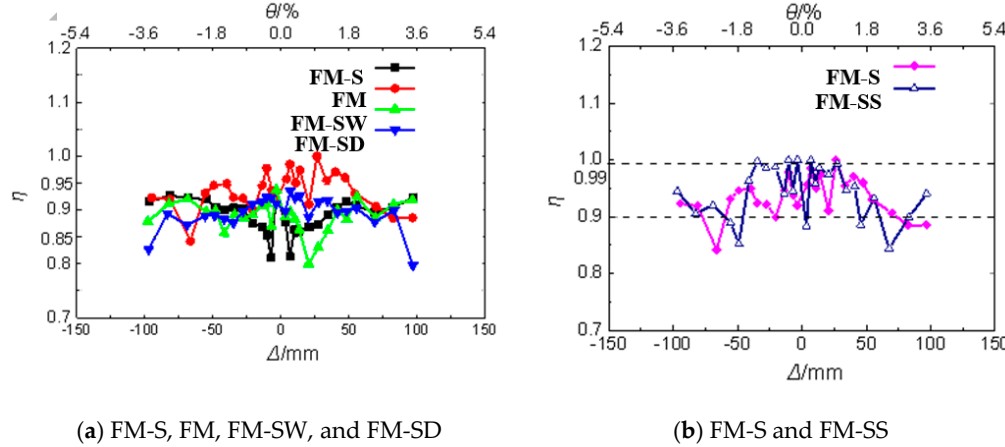

(**a**) FM-S, FM, FM-SW, and FM-SD          (**b**) FM-S and FM-SS

**Figure 18.** Variations in the strength deterioration factor with displacement angles of specimens.

Figure 18 showed that the patterns of the load-bearing capacity deterioration curves of the specimens were basically consistent, which indicates that the type of slab and the type of opening had an insignificant impact on the deterioration of the structural load-bearing capacity. The load-bearing capacity reduction factors, $\eta$, of the specimens under both positive and negative loading, were concentrated in the range of 0.85–0.95, which indicates that the structures retained a high residual load-bearing capacity after being subjected to cyclic loads. This characteristic contributed to a higher collapse-resistance performance of structures and facilitated post-seismic repair of the structures.

Figure 19 showed the secant stiffnesses $K_i$ of the specimens.

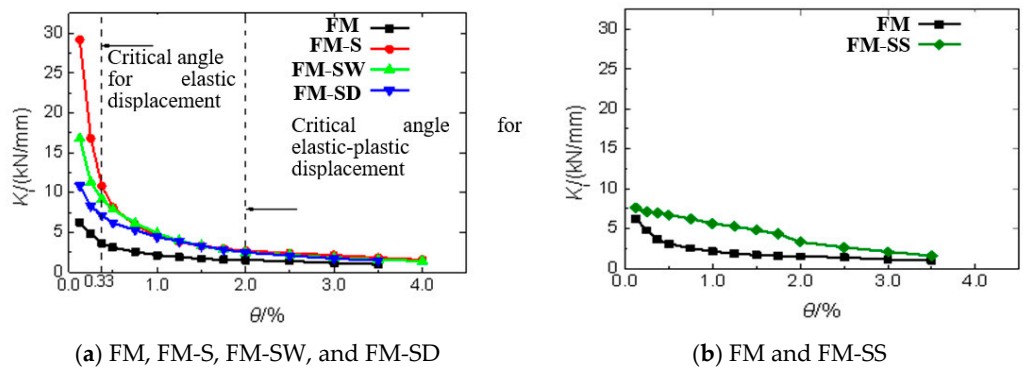

(**a**) FM, FM-S, FM-SW, and FM-SD          (**b**) FM and FM-SS

**Figure 19.** Secant stiffnesses of the specimens.

Figure 19 showed the order of the specimens in terms of the value of initial stiffness from high to low, which was given as: FM-S, FM-SW, FM-SD, FM-SS, and FM. The order of the specimens in terms of the stiffness reduction rate was the same. At displacement angles greater than 1%, specimens FM-S, FM-SW, FM-SD, and FM-SS exhibited basically consistent stiffness reduction patterns. Therefore, the 1% displacement angle could be used as a reference value for computing the elastic and plastic displacement angles when designing similar structures.

The major reason for the reduced lateral load-resistance stiffness of specimen FM was the deformation at the ends of the columns and beams of the lightweight CFST framework with the transition from elastic to elastic-plastic deformation, and the damage to the recycled concrete in the steel-tube columns. The stiffness reduction rate of the specimen FM was always smaller than the rates of specimens with sandwich slabs and concealed steel-truss slabs. The major reason was that the evolution of damages in the slabs was clearly faster than that in the lightweight CFST framework.

### 3.5. Energy Dissipation Capacity

For the purposes of this study, the equivalent viscous damping coefficient, $h_e$, and the accumulative dissipated energy, $E_i$, were employed to indicate the energy dissipation capacity of the specimens. Figure 20 showed variations in the equivalent viscous damping coefficient with the displacement angle of the tested specimens. Figure 21 showed the variations in accumulative dissipated energy with a displacement angle.

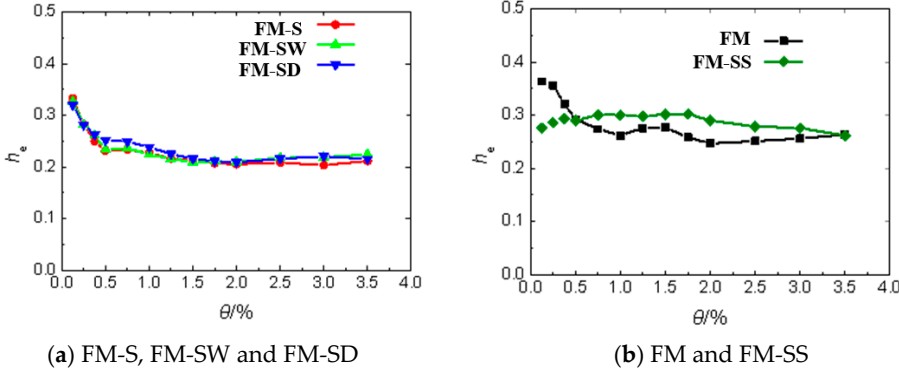

(**a**) FM-S, FM-SW and FM-SD　　　　　　　　　　(**b**) FM and FM-SS

**Figure 20.** Variations in the equivalent viscous damping coefficient with a displacement angle of the tested specimens.

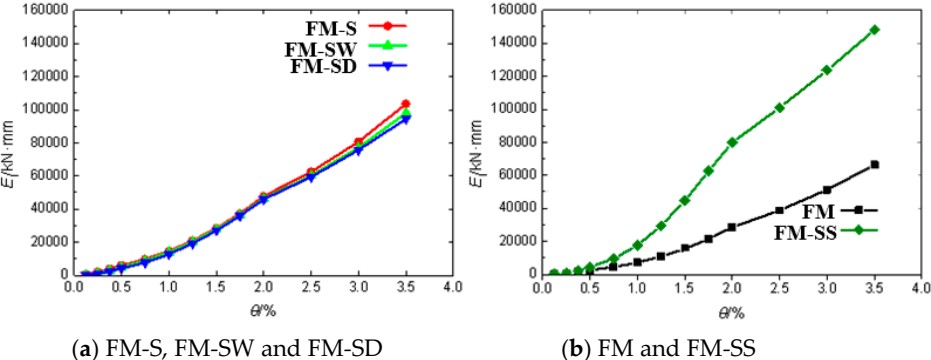

(**a**) FM-S, FM-SW and FM-SD　　　　　　　　　　(**b**) FM and FM-SS

**Figure 21.** Variations in accumulative dissipated energy with a displacement angle of the tested specimens.

Figures 20 and 21 revealed several things.

The equivalent viscous damping coefficients of all the specimens, except for specimen FM-SS, decreased rapidly at displacement angles smaller than 0.5%, decreased gradually at displacement angles from 0.5% to 2.0%, and exhibited an approximately flat, straight line at displacement angles greater than 2.0% until the failure of the specimens. The accumulative dissipated energies of specimens FM-S, FM-SW, and FM-SD were clearly greater than the accumulative dissipated energy of specimen FM. This was because sliding between the panels resulted in friction, and the friction resulted in energy dissipation, which contributed to a higher dissipated energy of the structure. This indicates that the sandwich slab and the framework acted synergistically by contributing to the greater energy dissipation capacity of the structure.

The variation patterns of the equivalent viscous damping coefficients of specimens FM-SW and FM-SD were the same as the pattern of specimen FM-S, and the accumulative dissipated energy of specimen FM-S was slightly greater than the accumulative dissipated energies of specimens FM-SW and FM-SD. This indicates that an opening to a sandwich slab had a degree of impact on the energy-dissipation capacity of the structure. Additionally, the energy-dissipation capacity of the specimen with a window opening was slightly greater than that of a specimen with a door opening.

The equivalent viscous damping coefficient of specimen FM-SS gradually increased at displacement angles smaller than 0.5%, remained at a stable level at displacement angles between 0.5% to 2.0%, and exhibited a decreasing trend at displacement angles greater than 2.0%. At displacement angles greater than 0.5%, the equivalent viscous damping coefficient and dissipated energy of specimen FM-SS were significantly greater than those of other specimens, which indicates that the concealed steel-truss slab and the lightweight CFST framework acted together to form a good energy dissipation system.

### 3.6. Analysis of Strains

Higher strains developed at the lower part of the framework columns and the ends of the specimen beam. Thus, an analysis of the cross-sectional strains in the lower part of the columns and at the ends of beams was performed on the typical specimens, FM, FM-S, and FM-SS. Figure 22 showed the cross-sectional lateral load-strain curves of the lower part of the columns of the typical specimens. The strains shown in the figure were measured at the middle of the steel-tube wall. Points 1 and 2 in the figure correspond to the yield loads, points 3 and 4 correspond to the ultimate loads, and points 5 and 6 correspond to the failure loads of the specimens.

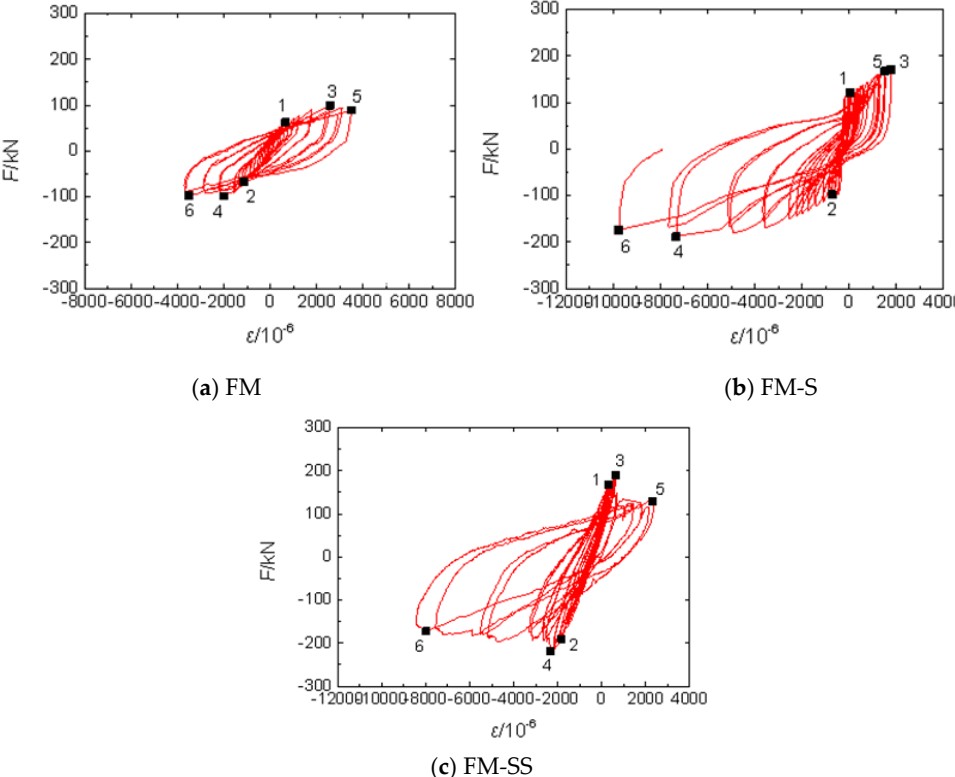

(**a**) FM　　　　　　　　　　　　　　　　　　　　(**b**) FM-S

(**c**) FM-SS

**Figure 22.** Cross-sectional lateral load-strain curves of the lower part of the columns of typical specimens.

Figure 22a,b showed that, before the specimens yielded, the steel tubes basically retained an elastic stage, which was followed by a stage when the strains in the steel tubes rapidly increased with the lateral load. The specimen then directly entered a plastic strengthening stage, exhibiting no clear yield point. More specifically, the compressive (−) and tensile (+) strains of specimen FM under the ultimate load were $2000 \times 10^{-6}$ and $2500 \times 10^{-6}$, respectively, whereas those of specimen FM-S were $7000 \times 10^{-6}$ and $2000 \times 10^{-6}$. This was because the steel tubes of specimen FM-S, supported by the sandwich slab, were capable of a greater deformation or were capable of bringing the greater deformability of steel into full effect.

Figure 22c showed that, under loads smaller than the yield load, specimen FM-SS exhibited an elastic strain. Under the yield load, the steel tubes exhibited a compressive strain close to $2500 \times 10^{-6}$. Under loads greater than the yield load, the compressive strain in the steel tube developed rapidly, because the columns of the framework acted as chords of the lightweight steel-truss at this stage and were, therefore, subjected to higher lateral loads.

Figure 23a–c showed the lateral load-strain curves at the ends of the framework beams. The strains shown in the figures were measured at the flange of the beams.

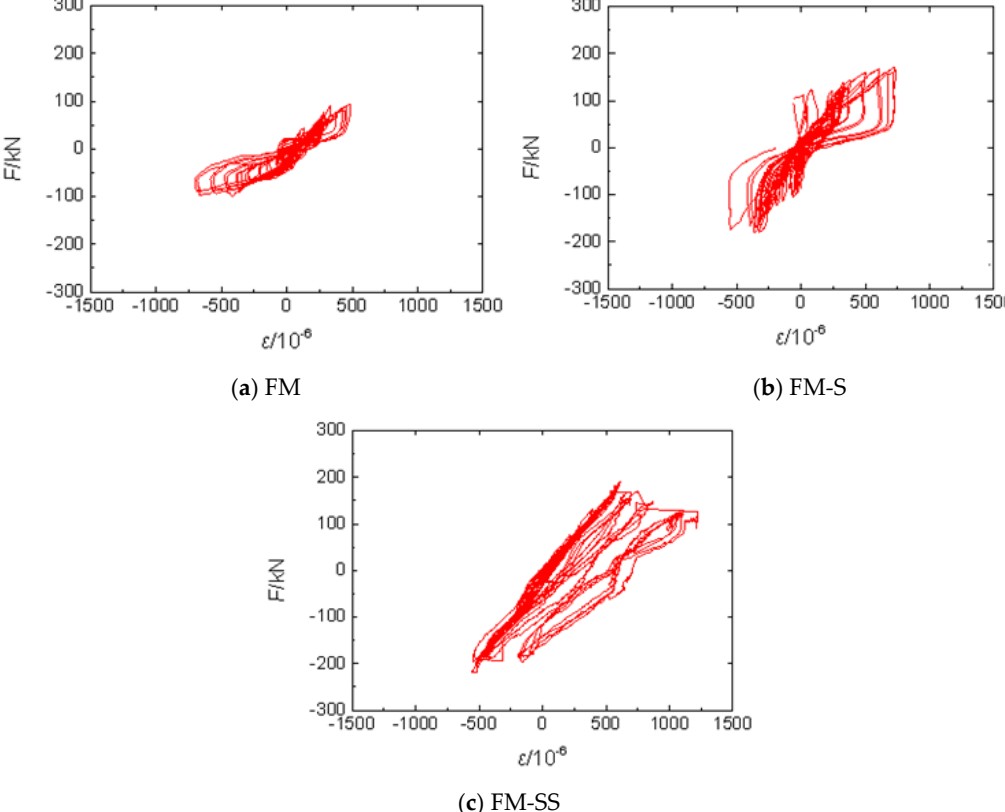

Figure 23. Lateral load-strain curves at the ends of the framework beams.

Figure 23 showed that, under the failure load, all strains of the specimens measured at the flange of the beams were smaller than the yield strain of $1500 \times 10^{-6}$. This was because the vertical load of the floor and slab on the framework beam acting on the steel H-beam were neglected for the test, and only the effect of the deformation of the framework columns on the steel H-beam was considered. This resulted in a smaller bending moment on the steel H-beam.

The typical specimen, FM, was selected to verify whether or not the cross-sectional strain in the lower part of the columns were consistent with the plane cross-section assumption. The cross-sectional strain in the framework columns could be expressed with reference to the coordinate system defined in Figure 24.

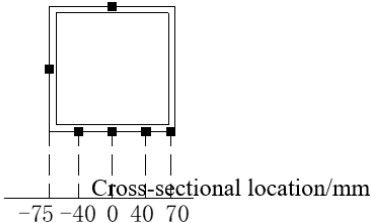

Figure 24. Cross-sectional location of the framework columns.

Figure 25 showed the distribution of the cross-sectional strain in the lower part of the columns. Under loads smaller than the yield load, the cross-sectional strain in the lower part of the framework columns of specimen FG were consistent with the plane cross-section assumption. Under loads higher than the yield load, the strain along the outer sides of the cross-section in the lower part of the framework column clearly increased, and the distribution of the cross-sectional strain was no longer consistent with the plane cross-section assumption.

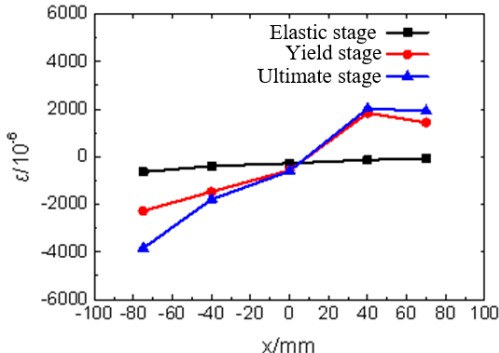

**Figure 25.** Distribution of the cross-sectional strain in the lower part of the columns.

## 4. Conclusions

(1) The prefabricated lightweight CFST framework and composite slab structure developed in this study had a good earthquake-resistance performance. The beam-column joint of the framework realized by a structure of double stiffener-reinforced L-brackets was reliable, which realized a framework of strong columns, a light beam, and stronger joints that enabled a ductile yield mechanism.

(2) The sandwich slab and framework worked in a well-coordinated manner when reasonably connected. Compared with an empty framework, one walled with sandwich panels had a clearly better lateral load-bearing capacity, stiffness, and deformability because the occurrence of lateral sliding and friction between sandwich panels contributed to a better energy dissipation capacity of the overall structure.

(3) A door or window opening in the slab contributed to a lower yield load, ultimate load, and initial stiffness of the overall structure, but had no clear impact on the accumulative dissipated energy of the overall structure. A larger opening in the slab resulted in a smaller yield load, ultimate load, and initial secant stiffness of the structure.

(4) The concealed steel-truss slab and framework worked in a well-coordinated manner. A concealed steel-truss slab contributed to a significantly higher ultimate load-bearing capacity of the framework and improved the lateral load-resistance performance of the structure.

**Author Contributions:** J.S. analyzed the data and wrote the manuscript. J.S. and C.W. designed the experiment. L.Z. performed the experiment. D.W. and S.Y. modified the paper.

**Funding:** "This research was funded by the National Natural Science Foundation of China (NO. 51508009) and the Fundamental Research Funds for the Central Universities of China (NO. 2652017078).

**Conflicts of Interest:** The authors declare no conflict of interest.

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
