# Peer review of "Experimental Study on a Prefabricated Lightweight Concrete-Filled Steel Tubular Framework Composite Slab Structure Subjected to Reversed Cyclic Loading"

_applsci, doi:10.3390/app9061264_

Round 1

Reviewer 1 Report

The paper shows an extensive experimental research.

The scientific originality is low but the experiments appear to be carried out correctly

The paper can be accepted for pubblication because the results of the experimental tests are in some way of interest and it can be useful to publish them, but the paper is poor by a scientific point of view. In someway it seems that the authors have tested an engineering project without a scientific point of view  

Author Response

Dear 

I upload the response as a word file 

Reviewer 2 Report

interesting paper and topic.

the suggestion is a "minor revision" of the paper, because there are few "major" requests only.

but please revise it, by taking into account all the following recommendations:

- general comment on the experiments: the examined system typology, and its performance under in-plane lateral loads, is very similar to timber log-house walls. The critical aspects of door/window openings, in this regard, has been largely studied - both numerically and experimentally - in order to derive some useful recommendations for their design. In this paper, the Authors should add some general comments and cite this 2 documents:

https://doi.org/10.1061/(ASCE)ST.1943-541X.0001065

https://doi.org/10.1016/j.engstruct.2015.03.002

There, the in-plane resisting mechanisms and failure mdoes are discussed. The key aspect of both these systems is the presence of structural panels asked to mechanically interact via a set of tongue-grooves joints. For both of them, moreover, the structural discontinuity due to the presence of door/window openings represent a critical aspect for safe / optima design. Please add some comments in the paper.

This kind of (qualitative) comparison will add value to the current discussion of results.

- following the previous comment: how the region of door/window openings could be improved / protected, with enhanced structural performances for the full specimens? Are the authors considering a possible (suitable) solution? please comment

- how damping / friction phenomena can modify / affect the observed experimental behaviours under in-plane lateral loads?

- how the presented results are sensitive to in-plane vertical loads? I can imagine that the tested walls are representative of a portion of a full building, so they are subjected to permanent and accidental loads that can range within a certain range (depending on the full building size). PLease comment and clarify the working assumptions for the experimental investigation, including few comments on possible limits / extension of these tests

- in the figures with drawings, please check that all the quotations have unit

- figure 7: figures (a) and (b) should be presented with a ruler / reference unit.This will allow the reader to understand their scale factor, for comparisons

- general comment for figures with multiple plots: the description of each figure item (a), (b),etc., must be explicitly included in the figure caption. All the figures must be revised, accordingly

-figures 9(a),(b) and (c) should be presented in the same size / scale factor. Why actually they have different size?

- tables 2 to 5: these material properties are nominal values? their source is not clear and must be clearly defined in the paper. Are these values from experiments? please comment and clarify

- are numerical simulations planned to extend this short experimental study? please comment in the paper

- extend the current list of literature references

Author Response

(The authors gave the same response as above.)

Reviewer 3 Report

The submitted manuscript presents an interesting experimental study on steel and concrete hybrid frames. The experimental campaign is well documented and provides original results to the best knowledge of this reviewer. However, it is opinion of this reviewer that the current version cannot be accepted and some improvements are suggested before an actual recommendation for publication can be made.

The introduction presents a quite general overview of problems related to sustainability of constructions in China.  After this general section, focus should be given to existing solutions on available structural solutions and recent research on steel structures with reinforced concrete infill walls, both conventional and innovative.  The current state of the art appears rather limited. This will help to more clearly identify the original contributions provided by this study and their significant in the civil engineering community.

Writings in Figure 2c and 9c are not clear. Please improve avoiding superposition of writings and drawings.

Not clear what the red elliptic mark is indicating in Figure 3. Please revise.

Author Response

(The authors gave the same response as above.)

Round 2

Reviewer 2 Report

.

Author Response

Dear reviewer:

Thank you for your comments, I have improved the paperplease see the blue label in the manuscript, which was the second revised.

Reviewer 3 Report

The Authors have improved their manuscript. However, the review of the state of the art on steel frames with reinforced concrete infill walls is still very poor. This is not acceptable today, given the many tools for bibliographic research available on the internet. Just go to Scopus or Web of Science, type “seismic steel frames with reinforced concrete infill walls” and dozens of journal articles will be displayed. It is important that the Authors comment on the existing publications to clearly highlight their original contribution. Otherwise, the submitted manuscript cannot be recommended for publication.

Author Response

Dear reviewer:

Thank you for your comments, I have improved the paper,please see the blue label in the manuscript, which was the second revised.